# Antisense Oligonucleotides against Let-7 Enhance the Therapeutic Potential of Mesenchymal Stromal Cells

**DOI:** 10.3390/ijms24108639

**Published:** 2023-05-12

**Authors:** Dae-Won Lee, Sungho Shin, Jeong-Ho Kim, Cheolju Lee, In Yong Kim, Il-Hoan Oh

**Affiliations:** 1Catholic High-Performance Cell Therapy Center & Department of Medical Life Science, College of Medicine, The Catholic University, Seoul 06591, Republic of Korea; lovebauoo@catholic.ac.kr (D.-W.L.); inyongkim79@gmail.com (I.Y.K.); 2Chemical & Biological Integrative Research Center, Korea Institute of Science and Technology, Seoul 02792, Republic of Korea; sungho@kist.re.kr (S.S.); clee270@kist.re.kr (C.L.); 3KHU-KIST Department of Converging Science and Technology, Kyung Hee University, Seoul 02447, Republic of Korea; 4Regen Innopharm Inc., Seoul 06591, Republic of Korea; leeluleel@catholic.ac.kr

**Keywords:** mesenchymal stromal cells, LIN28, Let-7, ASO, therapeutic potential

## Abstract

Let-7 miRNAs have pleiotropic cellular functions in cell proliferation, migration, and regenerative processes. Here, we investigate whether the inhibition of let-7 miRNAs with antisense oligonucleotides (ASOs) can be a transient and safe strategy enhancing the therapeutic potential of mesenchymal stromal cells (MSCs) to overcome their limitations in cell therapeutic trials. We first identified major subfamilies of let-7 miRNAs preferentially expressed in MSCs, and efficient ASO combinations against these selected subfamilies that mimic the effects of LIN28 activation. When let-7 miRNAs were inhibited with an ASO combination (anti-let7-ASOs), MSCs exhibited higher proliferation with delayed senescence during the passaging into a culture. They also exhibited increased migration and enhanced osteogenic differentiation potential. However, these changes in MSCs were not accompanied by cell-fate changes into pericytes or the additional acquisition of stemness, but instead occurred as functional changes accompanied by changes in proteomics. Interestingly, MSCs with let-7 inhibition exhibited metabolic reprogramming characterized by an enhanced glycolytic pathway, decreased reactive oxygen species, and lower transmembrane potential in mitochondria. Moreover, let-7-inhibited MSCs promoted the self-renewal of neighboring hematopoietic progenitor cells, and enhanced capillary formation in endothelial cells. These findings together show that our optimized ASO combination efficiently reprograms the MSC functional state, allowing for more efficient MSC cell therapy.

## 1. Introduction

Mesenchymal stromal cells (MSCs) are a non-hematopoietic adherent cell population derived from the bone marrow (BM), adipose tissue, or placental tissue. MSCs exhibit multi-lineage differentiation potential towards diverse types of tissue, including bone, cartilage, and adipose [1,2,3]. MSCs have been obtained from a variety of sources, including BM, umbilical-cord blood, adipose tissues, or the placenta [4]. Accumulating studies have shown that the primary mode of action of MSCs is the paracrine support of tissue regeneration via inhibiting apoptosis and fibrosis [5], and stimulating the regeneration of endogenous stem cells such as hematopoietic stem cells (HSCs), neuronal stem cells, and other tissue-specific endogenous stem cells [6,7]. So far, more than 425 clinical trials have been performed using ex vivo-expanded MSCs to facilitate the regeneration of injured tissues [8].

Variability has been observed in MSC-based clinical trials, and recent meta-analyses on the efficacy of MSC-based cell therapy in rigorously designed clinical studies indicates a lack of significant efficacy in patients with cerebral stroke [9], myocardial infarction [10], and in clinical trials to suppress graft vs. host immune reactions, thus posing a dilemma in the ongoing use of MSCs [11]. Accumulated studies have identified many factors that limit the therapeutic potential of MSC-based cell therapy, including variations among donor cells and functional discrepancies generated during the in vitro culture of MSCs [12,13,14]. In particular, MSCs obtained from primary tissues exhibited limited proliferative capability and replicative senescence after in vitro culture [15], characterized by a reduction in proliferation rate, enhanced mitochondrial reactive oxygen species (ROS) generation, telomere shortening (or DNA damage), and the impairment of functional properties [16]. Accordingly, strategies to increase their growth capability and regenerative potential are of major interest in the field.

The let-7 family of miRNAs contains multiple members that are highly conserved in animals. They have pleiotropic functions targeting a diverse spectrum of mRNAs, including genes regulating cell proliferation (HMGA1, c-MYC, RAS, HMGA2) [17,18,19], cell cycle progression (CyclinD1, CDC34) [20], angiogenesis (VEGF, TGFBR) [21,22], and glucose metabolism (IGF1R, INSR, IRS2) [23]. Moreover, let-7 miRNAs participate in cellular processes controlling cell fate, stemness, or oncogenesis [24,25].

The biogenesis of let-7 is suppressed by RNA-binding protein LIN28, where LIN28 binds to the terminal loop of let-7 precursors in the conserved motif GGAG to inhibit the molecular interaction of let-7 miRNA with Dicer, thereby inhibiting the maturation of let-7 miRNA by Dicer [26,27]. Accordingly, a reciprocal relationship was observed between the expression levels of LIN28 and let-7 miRNA, where LIN28 is highly expressed in stem cells or cells in embryogenesis, and mature let-7 levels are increased during stem cell differentiation or during aging [23]. Therefore, the downregulation of let-7 miRNA is associated with the upregulation of LIN28, which promotes the regeneration of mesenchymal tissue after injury [23], the proliferation of hematopoietic stem cells [28], or the expansion of neural crest progenitors [29]. Accordingly, there is a possibility that the inhibition of the let-7 miRNA should mimic the effects of LIN28 to enhance the regenerative activity of stem cells. However, the extent to which the inhibition of let-7 miRNA could mimic the pro-regenerative effects of LIN28 is not yet clear, since let-7 is not the only biological target of LIN28 [30].

Antisense oligonucleotides (ASO) against miRNAs can inhibit the biological function of miRNAs by either blocking the association of miRNA and mRNA transcripts or promoting the degradation of the target miRNAs [31,32]. These ASOs, which have short half-lives, have been safely utilized in large numbers of clinical trials against somatic and genetic diseases [33].

Therefore, in this study, we address whether the ASO-based inhibition of let-7 miRNAs could enhance the regenerative potential of MSCs. We found that specific sets of ASOs against let-7 miRNAs could efficiently knock down the majority of let-7 miRNA families, promoting MSC proliferation and regeneration. We demonstrate the feasibility of using ASO therapy to promote higher MSC therapeutic potential.

## 2. Results

### 2.1. Inhibition of Let-7 Expression in MSCs

Since there are multiple subfamilies of let-7 miRNAs, we first analyzed the relative abundance of let-7 miRNA subtypes to identify the major subfamily genes being preferentially expressed in MSCs. Thus, the expression level of each let-7 miRNA subtype in human adipose-tissue-derived MSCs (AD-MSCs) were analyzed by comparing the reads per kilobase of transcript per million mapped reads (RPKMs) in RNA-seq data. As shown in Figure 1A, let-7a-5p was the most abundantly expressed in AD-MSCs, followed by let-7b-5p (1.6 × 10^5^ and 2.3 × 10^4^ RPKMs, respectively), while the levels of other let-7 miRNA subfamilies (let-7c–g,i) were significantly lower (<5 × 10^3^ RPKMs). Similarly, the RQ-PCR analysis of the let-7 miRNA subfamily in BM-derived MSCs (BM-MSCs) obtained from three independent donors revealed a predominantly higher expression of let-7a-5p miRNAs than that of other subtypes of let-7 miRNAs (Figure 1B). These results indicated that let-7a-5p is the major subtype of let-7 miRNAs predominantly expressed in the MSCs. Therefore, we chose an ASO mixture targeting let-7a and let-7b to investigate the effects of inhibiting let-7 miRNA in MSCs on the basis of RNA-seq analysis showing let-7b-5p as the second most abundant subtype, and miRWalk DB-based bioinformatic analysis, where target genes of let-7a and let-7b cover the majority of the total target genes of let-7 miRNAs (91% of total let-7 family target genes), primarily due to the conserved seeding region sequences of the let-7 family (Figure 1C,D). To confirm the significance of selecting these subfamilies, we transfected a mixture of ASOs against let-7a and let-7b, and found that most of the let-7 family levels (let-7a, b, d, e, f, i) were significantly inhibited by this ASO combination (referred to as anti-let7-ASO) (Figure 1E), thus indicating that the selected ASOs against let-7a-5p and let-7b-5p could inhibit most of the families of let-7 miRNAs.

Next, we examined the functional consequences of let-7 inhibition in MSCs by transfecting anti-let7-ASO into MSCs. We found a significant induction of HMGA2, IGF2BP1, and IGF2BP2, the representative target genes of let-7 miRNAs [17,18,19] (Figure 1F). Mature let-7 itself could bind to the 3′ untranslated region of LIN28 mRNA to cause the post-transcriptional repression of LIN28 [34]. Therefore, we also examined the expression levels of LIN28 and found a significant upregulation of the LIN28 protein level (Figure 1G,H). These results confirm that the anti-let7-ASO could be efficiently utilized as a functional inhibitor of let-7 miRNAs in MSCs.

### 2.2. Functional Effects of Let-7 Inhibition in MSCs

Having validated the inhibitory effects of anti-let7-ASOs, we examined the influence of let-7 inhibition on autonomous MSC functions. The effects of anti-let7-ASO were first evaluated on the proliferation of MSCs. We found that the target let-7 miRNA genes and cell-cycle regulators c-MYC, CDC34, and CCND2 were significantly induced in MSCs (Figure 2A). Accordingly, anti-let7-ASO increased the in vitro proliferation of MSCs, leading to higher expansion folds (Figure 2B,C) with lower levels of senescence-associated β-galactosidase activity than those of control MSCs (Figure 2D,E). However, despite these differences in proliferative potential, telomeric length did not differ between the MSC groups (Figure 2F). These results together indicate that the inhibition of let-7 by anti-let7-ASO could enhance ex vivo proliferative potential and reduce senescence without changes in telomeric length.

We next examined the influences of let-7 inhibition on multi-lineage differentiation potential and phenotypic changes in MSCs. When subjected to osteogenic or adipogenic differentiation, let-7-inhibited MSCs exhibited higher osteogenic potential, but comparable levels of adipogenic differentiation (Figure 3A–C). Previous studies showed that more primitive populations of MSCs or ontologically earlier stages of MSCs proliferate more and have enhanced osteogenic potential [35,36]. Therefore, taking this similarity in the functional changes into account, we investigated whether let-7-inhibited MSCs acquire stemness by examining changes in the frequencies of colony forming cells (CFU-F) and the expression of pericyte markers [37,38,39] or stemness-related genes. However, we did not find significant increases in the frequency of CFU-F, the expression of pericyte markers (NG2, NESTIN, PDGFR-β), or the expression of stemness-related genes (OCT4, SOX2, NANOG, and KLF4) (Figure 3D–G). These results show that the functional changes induced by anti-let7-ASO were not changes in cell fate, but rather reflect differences in cellular functional state.

Interestingly, the let-7-inhibited MSCs exhibited a profound increase in cell migration, consistent with the induction of HMGA2 and IGF2BP2, genes regulating cell migratory activities [40,41] (Figure 4A,B). However, this increase in cell migration was not associated with an increase in epithelial–mesenchymal transition (EMT)-regulating factors, since there were comparable levels of SNAI1, SLUG, ZEB1, ZEB2, and TWIST (Figure 4C) similarly suggesting functional changes without alterations in cell fate.

Next, we were interested to see whether effects from let-7 inhibition were analogous to the effects of LIN28 overexpression. In addition, considering that let-7 is not the exclusive downstream target of LIN28 [30], we were prompted to see to what extent the effects of let-7 inhibition mirror the overexpression of LIN28 in MSCs. As shown, the overexpression of the LIN28 protein caused a similar downregulation of let-7 family miRNAs and the induction of target genes (HMGA2) (Appendix A and Figure 5A,B). Moreover, these LIN28-overexpressing MSCs exhibited a similar increase in migration and osteogenic differentiation without changes in adipogenic differentiation, and comparable frequencies of CFU-F, thus indicating similar functional changes as from let-7 inhibition (Figure 5C–H). Likewise, there were no significant difference in the expression of EMT-related genes between the LIN28-overexpressed and control MSCs (Appendix A). However, LIN28 overexpression did not cause an increase in cell proliferation, unlike the effects from let-7 inhibition, thus indicating a similar but distinct biological outcome to that of let-7 inhibition (Figure 5I). Together, these results indicate that the inhibition of let-7 mimics the functional changes induced by LIN28 overexpression, but with partial discrepancies, indicating an unique functional influences on MSCs via let-7 inhibition.

Given that MSCs undergo functional changes without differences in cell fate, we investigated the possibility that MSCs with let-7 inhibition exhibit differences in metabolic regulation, as previously observed for LIN28 overexpression [23]. First, we examined the influence of let-7 inhibition on the glucose metabolism. The let-7-inhibited MSCs had higher expression level of genes involved in glucose uptake, including GLUT1, IGF1R, and INSR (Figure 6A), and enhanced glycolytic capacity with a higher extracellular acidification rate (ECAR) (Figure 6B,C). In contrast, let-7-inhibited MSCs exhibited lower ROS levels compared to those of the controls, indicating that the glucose metabolism was reprogrammed from oxidative phosphorylation to anaerobic glycolytic pathways (Figure 6D,E). Supporting this possibility, the mitochondrial membrane potential, as measured with TMRE staining, was significantly lower in let-7-inhibited MSCs, indicative of a decrease in oxidative phosphorylation [42] (Figure 6F,G).

Taken together, these results indicate that let-7 inhibition in MSCs leads to metabolic reprogramming characterized by lower ROS levels and lower mitochondrial activity, mimicking the metabolic characteristics of stem cells [43].

### 2.3. Proteomics of Let-7-Inhibited MSCs

To further characterize functional MSC changes induced by let-7 miRNA inhibition, we examined changes in the proteomics in the MSCs, taking the post-transcriptional regulatory effects of miRNAs into account to influence the protein levels [31,32]. We compared global protein expression patterns in anti-let7-ASO-transfected and control MSCs via LC–MS/MS analysis. Through comparing the proteomes of let-7-inhibited and control MSCs, we identified 262 proteins induced in the anti-let7-ASO-transfected MSCs (Appendix A), of which HMGA2 and IGF2BP2, the target genes of let-7, were the most profoundly induced proteins (Appendix A). The gene ontology (GO) analysis of significantly induced proteins (>1.3 fold) in let-7-inhibited MSCs showed an enrichment of proteins involved in multiple cellular functions, including wound healing, angiogenesis, cell migration, and the energy metabolism (Figure 7A). Similarly, ingenuity pathway analysis (IPA) showed the activation of cell migration and cell viability, and the inhibition of apoptosis (Figure 7B). Thus, proteomic changes induced by the inhibition of let-7 miRNAs were consistent with functional changes, including enhanced migration and higher proliferative potential.

### 2.4. Higher Cell Therapeutic Potential of Let-7-Inhibited MSCs

Having observed the augmentation of multiple cell-autonomous functions of MSCs by let-7 inhibition, we investigated the paracrine supportive functions of MSCs on their neighboring cells. We first examined the ability of MSCs to support hematopoietic progenitor cells (HPCs) by serving as a niche for HPCs [7,44]. Therefore, we co-cultured anti-let7-ASO or control-transfected MSCs with CD34^+^ cells from human umbilical-cord blood, and examined the expansion of primitive subsets of hematopoietic progenitors (CD34^+^ CD90^+^) which have multi-lineage reconstitution potential [45] (Figure 8A). As shown, compared to the control-group MSCs, MSCs transfected with anti-let7-ASO exhibited a higher expansion of CD34^+^ CD90^+^ cells (Figure 8B). In contrast, these supportive effects of let-7-inhibited MSCs were not observed for leukemia cells (Appendix A), indicating their selective support of normal HPCs.

We also examined the effects of let-7 inhibition in MSCs on the angiogenesis of endothelial cells using human umbilical-vein endothelial cells (HUVEC) co-cultured with each group of MSCs (Figure 8C). The level of capillary formation by HUVECs, as quantified by the numbers of nodes, junctions, and meshes, was significantly increased in the group co-cultured with let-7-inhibited MSCs compared to those cultured with control MSCs (Figure 8D,E). Similarly, let-7-inhibited MSCs had accelerated wound healing compared to the controls, consistent with the proteomic data that showed the upregulation of pathways for wound healing (Appendix A). Together, these results show that anti-let7-ASO could enhance the supportive function of MSCs on the self-renewing expansion of HPCs, the angiogenesis of endothelial cells, and wound healing, indicating the higher regenerative potential of the cells.

## 3. Discussion

Suboptimal clinical outcomes have been the major hurdles in MSC-based cell therapeutic trials. Limitations originate from multiple factors, including intrinsic cellular heterogeneity or limited numbers of cells during ex vivo culture. For example, significant functional heterogeneity exists in the ability of MSC subsets to stimulate tissue-specific stem cells, where only a primitive subset of MSCs can serve as stem-cell-supporting niche cells in vivo [37,38,39,46,47,48] and in vitro [49]. In addition, ex vivo cultured MSCs exhibit functional and molecular discrepancies to the in vivo naïve state [12,13,14], posing other challenges in MSC-based cell therapeutic trials. Accordingly, strategies to enhance the ex vivo expansion of MSCs and their regeneration-supporting potential are of major interest in a diverse spectrum of clinical situations [50].

In this light, we investigated the possibility that the regeneration-supporting activity of MSCs is enhanced by the ASO-mediated inhibition of let-7 miRNAs, taking the regeneration-promoting effects of LIN28 into account. In particular, considering that ASO therapies have transient action and a short half-life [33], we postulated that anti-let7-ASO could be a safe strategy to enhance the regenerative functions of MSCs. To pursue this possibility, we first analyzed the relative expression of multiple families of let-7 miRNAs in MSCs, and identified let-7a and let-7b as major subfamilies of let-7 miRNAs in AD-MSCs or BM-MSCs via both expression analysis and bioinformatics analysis using miRWalk DB. Importantly, we found that ASOs against these two let-7 miRNA subfamilies could efficiently downregulate the majority of the let-7 family miRNAs that we tested, probably due to highly conserved seed sequences within the group.

The inhibitory effects of these anti-let7-ASOs were verified through the significant de-repression of let-7 target genes in transfected MSCs, including HMGA2, IGF2BP1, or IGF2BP2, and cell-cycle regulating genes such as c-MYC, CDC34, or CCND2 [17,18,19]. Moreover, the upregulation of LIN28 proteins was also observed, reflecting the negative controlling effects of let-7 on LIN28 expression [34]. Accordingly, the ASO combinations that we optimized in this study caused the efficient inhibition of let-7 miRNAs in MSCs.

When MSCs were inhibited with let-7 miRNAs, the immediate difference that we observed in cell autonomous function was enhanced proliferation and osteogenic activity. Considering that primitive MSC populations or MSCs of ontologically earlier stages exhibit similar findings [35,36], it was possible to speculate that let-7 inhibition in MSCs acquired a cell fate mimicking the primitive populations of MSCs. However, in our study, the increment in the frequencies of CFU-F was not observed in let-7-inhibited MSCs or in LIN28-overexpressed MSCs. In addition, the expression levels of stemness-related genes (KLF4, OCT4, NANOG, SOX2) or pericyte markers (NG2, NESTIN, PDGFR-β) were not increased in let-7-inhibited MSCs, indicating that the functional changes in let-7-inhibited MSCs were not related to the cell fate changes. Let-7-inhibited MSCs also exhibited significantly enhanced migration potential without expression changes in the genes controlling EMT, the process controlling cellular interaction with the extracellular matrix and movement of cells [51]. Thus, the changes in the biological properties of let-7-inhibited MSCs appeared to occur in the metabolic and functional levels, rather than being driven by cell fate changes into more primitive or more mesenchymal phenotypes.

Consistent with the findings, let-7-inhibited MSCs exhibited characteristics of metabolic reprogramming, i.e., anti-let7-ASO-treated MSCs exhibited higher glucose consumption and enhanced glycolytic capacity, as shown by enhanced ECAR. Moreover, let-7-inhibited MSCs exhibited lower ROS levels and a significant decrease in mitochondrial transmembrane potential. These findings suggest that ASO-mediated let-7 inhibition leads to the metabolic reprogramming of MSCs, shifting their balance from oxidative phosphorylation to anaerobic glycolytic pathways with lower mitochondrial activity, which is reminiscent of the findings from LIN28 activation in stem cells and embryonic kidney cells [42,52].

Importantly, the functional changes induced by anti-let7 ASO may enhance MSC therapeutic potential. First, the MSCs exhibited higher proliferative potential with a delay in senescence, as shown by higher ex vivo expansion and lower levels of senescence-associated β-galactosidase activity. Since the current cell therapeutic trials are based on large-scale ex vivo expansion but efficient expansion is limited by senescence, increasing the long-term expansion potential of MSCs could be advantageous for the larger-scale preparation of MSCs.

Second, let-7-inhibited MSCs exhibited an enhanced ability to support neighboring cells, i.e., they promoted the self-renewal/maintenance of neighboring hematopoietic progenitors, as evidenced by the higher expansion of CD34^+^ CD90^+^ cells, the hematopoietic progenitor subset representing primitive long-term repopulating cells [45]. However, there is the possibility that these stimulating effects of MSCs promote the growth of tumor cells or their migration, as recently suggested by a study on the prostate-cancer-associated MSCs and their secretion of IL-6, the target of let-7 miRNAs [53]. Considering the cell-type-specific effects of MSCs on various cancer cells, a series of studies on this possibility are warranted.

Lastly, MSCs treated with anti-let7-ASO exhibited increased migratory activity. The migration of MSCs is important for the recruitment of MSCs to the injury site and for regeneration. The enhanced migration potential was also evident in proteomic analysis, where proteins upregulated by anti-let7-ASO were highly enriched in those related to the migration and viability of the cells. Moreover, let-7-inhibited MSCs promoted angiogenesis in endothelial cells, which could facilitate the regeneration of injured tissues. Accordingly, let-7-inhibited MSCs caused a significant acceleration of wound-healing activity, suggestive of their enhanced tissue regeneration potential.

Collectively, the inhibition of let-7 miRNA in ASO-based platform could be an attractive cell-manipulating approach by virtue of their transient and short half-life, and their transient microenvironment-stimulating effects on regenerating cells without the safety issues related to the stable genetic modification of the cells.

Taken together, our results show that our optimized anti-let7 ASOs efficiently inhibited multiple families of let-7 miRNAs and enhanced the therapeutic potential of MSCs. Further evidence, however, is warranted to pursue this possibility in various disease conditions. Nevertheless, our results raise the possibility that anti-let7 ASOs could be utilized for safer and more efficient cell therapeutic strategies in MSC-based cell therapy for better therapeutic outcomes.

## 4. Materials and Methods

### 4.1. Cell Culture

Cells in umbilical-cord blood were obtained from the Seoul Metropolitan Government Public Cord Blood Bank. BM-MSCs from BM aspirates were obtained from healthy donors after their informed consent under approval from the Institutional Review Board of the Catholic University of Korea and Seoul St. Mary’s Hospital, the Catholic University of Korea (MC19TNSI0012 and MC19SNSI0059). MSC cultures were established from the mononuclear cell fraction and maintained in low Dulbecco’s modified Eagle’s medium (Cytiva, Uppsala, Sweden) containing 10% fetal bovine serum (Cytiva), L-glutamine (Gibco, Waltham, MA, USA), antibiotics (Anti-Anti^TM^, Gibco) in a humidified 5% CO_2_ atmosphere at 37 °C.

Human umbilical-vein endothelial cells (HUVECs, Lonza, Walkersville, MD, USA) were cultured in complete EGM-2 medium (Lonza, Basel, Switzerland) according to the manufacturer’s instructions. Cells were incubated at 37 °C with 5% CO_2_ and maintained using standard cell culture techniques.

HL60 and MV-4-11 were cultured in an IMDM growth medium (Gibco) according to the ATCC handling information. Similarly, MOLM14 was cultured in an RPMI medium (Gibco) with 10% FBS according to the DSMZ information.

### 4.2. Transfection of ASO against miRNAs

ASOs for control and miRNA inhibitors were purchased from Bionics Corp. (Daejeon, Korea). ASOs were tagged with 3′ cholesterol and modified with 2-O-methylation and phosphorothioate. BM-MSCs were cultured in growth media without Anti-Anti^TM^ and transfected with 10 nM miRNA in Opti-MEM (Gibco) using a Lipofectamine RNAiMAX Transfection Reagent (Thermofisher, Waltham, MA, USA).

### 4.3. RNA Extraction and Quantitative RT-PCR

Total RNA from MSCs was isolated with Trizol (Invitrogen, San Diego, CA, USA). cDNA was synthesized from l–2 µg of total RNA with SuperiorScript III Mastermix (Enzynomics, Daejeon, Korea). mRNA levels were measured via quantitative real-time PCR (qRT-PCR) using SYBR Premix Ex Taq II (Takara, Japan). Normalization and fold changes were calculated using the 2−∆∆Ct method, using HPRT as an internal control. The primer sequences used in this study are listed in Appendix A. For the analysis of miRNA expression, we used the miScript II RT Kit (QIAGEN, Hilden, Germany) and miScript SYBR^®^ Green PCR Kit (QIAGEN). For the normalization of miRNA levels, the expression levels of U6 snRNA in the cells were used as an internal control.

### 4.4. SA-β-Galactosidase Assay

The senescence-associated-β-galactosidase assay was performed using a senescence β-galactosidase staining kit (Cell Signaling Technology, Danvers, MA, USA) on late-passage cultured cells according to the manufacturer’s protocol.

### 4.5. Telomeric Length Analysis

The telomeric length of MSCs was measured with quantitative PCR to compare the telomere repeat sequence copy number relative to the single-copy gene (36B4) copy number. Telomere-specific primers (forward: 5′GGTTTGTTTGGGTTTGGGTTTGGGTTTGGGTTTGGGTT3′; reverse: 5′GGCTTGCCTTACCCTTACCCTTACCCTTACCCTTACCCT3′) and 36 B4 primers (forward: 5′CAGCAAGTGGGAAGGTGTAATCC3′; reverse: 5′CCCATTCTATCATCAACGGGTACAA3′) were prepared. All PCRs were performed on the Rotor-Gene Q real-time instrument (QIAGEN). Relative telomeric length was determined via the copy numbers corresponding to the T (telomere) and S (single copy; b-globin) amounts in each sample to calculate the relative T/S ratios.

### 4.6. Extracellular Metabolic Flux Analysis

Cultured cells were seeded onto XF 24 well Culture Microplates (Seahorse Bioscience, Lexington, MA, USA) at a density of 2 × 10^4^ cells/well and incubated overnight. The extracellular acidification rate (ECAR) was measured using an XF24 Extracellular Flux Analyzer (Seahorse Bioscience) according to the manufacturer’s protocol. ECAR were normalized to the cell numbers. For ECAR analysis, cells were sequentially treated with 10 mM glucose, 1 μM oligomycin, and 50 mM 2-deoxy-D-glucose (2-DG).

### 4.7. Mitochondrial Transmembrane Potential and ROS Measurement

MSCs treated with miRNA inhibitors were plated in 6-well cell culture plates. To detect the ROS levels, cells were washed with PBS and incubated in 20 nM DCF-DA in 2% FBS + HBSS media for 45 min. To measure the mitochondrial membrane potential, we used a TMRE-Mitochondrial Membrane Potential Assay Kit (Abcam, Cambridge, UK) according to the manufacturer’s protocol. In both experiments, we took images with a Lionheart FX and analyzed them with the GEN5 program.

### 4.8. Colony Formation and Multilineage Differentiation of MSCs

For the colony formation of MSCs (CFU-F), MSCs were plated in a dish (1000 and 2000 cells per 100 mm dish) and incubated for 14 days; then, the colonies containing >50 cells were counted by staining with crystal violet (Sigma-Aldrich, Carlsbad, CA, USA). Osteogenic differentiation was induced by a STEMPRO^®^ Osteogenesis Differentiation Kit (Gibco). After 12 days, the mineralization of the extracellular matrix was determined via Alizarin Red S staining and the extraction with 10% cetylpyridinium chloride, followed by measuring absorbance at 570 nm. Adipogenic differentiation was induced with a STEMPRO^®^ Adipogenesis Differentiation Kit (Gibco). After 12 days, cells were fixed with propylene glycol and stained with Oil Red O to visualize the lipid droplets. The numbers of lipid droplets were subsequently determined with dye extraction using 4% Nonidet P40 in isopropyl alcohol, followed by spectrophotometry at 510 nm.

### 4.9. Cell Migration Assay

The scratch assay was performed by plating 6 × 10^4^ cells on a 6-well plate. After 2 days of transfection, a sterile 200 μL pipette tip was used to create a straight scratch across the monolayer. The dish was washed to remove any detached cells and debris, and a fresh culture medium was added. Photographs were taken at 0 and 18 h using a microscope, and the images were analyzed using ImageJ to calculate the area of the scratch that was unoccupied by cells. Cell migration was quantified by measuring the change in the unoccupied area between 0 and 18 h using the following formula: (empty area at 18 h − empty area at 0 h)/empty area at 0 h.

### 4.10. Co-Culture with MSCs

Human CD34^+^ cells were purified from umbilical-cord blood using a CD34 MicroBead kit (Miltenyi Biotec, NRW, Bergisch Gladbach, Germany). For the co-culture, MSCs were irradiated (15 Gy) 24 h before the co-culture, washed, and co-cultured with purified CD34^+^ cells for 4 days in DMEM + 10% FBS (Cytiva, Marlboroug, MA, USA) in the presence of cytokine mixtures containing 20 ng/mL hSCF, 20 ng/mL hFlt3L, 4 ng/mL hIL-3, 4 ng/mL hIL-6, 4 ng/mL hG-CSF, and 0.2 × 10^6^ M hydrocortisone (ProSpec-Tany TechnoGene Ltd., Ness-Ziona, Israel). For the phenotypic analysis of ex vivo expanded hematopoietic progenitor cells, co-cultured cells were stained with antibodies against CD45, CD34, and CD90 (BD Pharmingen, NJ, USA), and analyzed with flow cytometry after gating the hematopoietic (CD45^+^) population. For the co-culture with the leukemia cell lines, MSCs were irradiated (15 Gy) and co-cultured with leukemia cell lines for 4 days in RPMI growth media (MOLM14) or IMDM growth media (HL60, MV-4–11). For the phenotypic analysis of leukemic cells, co-cultured cells were stained with antibodies against CD45 and CD90 (BD Pharmingen), and analyzed wih flow cytometry after gating the hematopoietic (CD45^+^) population.

HUVECs were co-cultured with MSCs for 3 days and sort-purified for HUVECs (CD90^−^) using BD FACSAria™ Fusion Flow Cytometers. The purified HUVEC cells were then loaded onto growth factor-reduced Matrigel (Corning, NY, USA), which comprised EBM-2^TM^ and reduced amounts of FBS and VEGF (1/4 and 1/2, respectively) of the stocks provided in the 1 × EGM^TM^-2 SingleQuots^TM^ Supplement Pack (Lonza, Basel, Switzerland).

### 4.11. Western Blot Assay

Cells were lysed in Np40 buffer (Pierce, Appleton, WA, USA). Proteins were subjected to SDS-PAGE gel and transferred to a nitrocellulose membrane (GE Healthcare, Chicago, IL, USA). The membrane was blocked for 1 h in PBST containing 5% milk, and subsequently probed with primary antibody against LIN28 (Abcam, ab124765) overnight at 4 °C. After 1 h incubation with donkey-anti-rabbit HRP-conjugated secondary antibody (GE Healthcare, Chicago, IL, USA), the protein level was detected using SuperSignal West Femto Luminol reagents (Invitrogen, Waltham, MA, USA).

### 4.12. Lentivirus Construction and Transfection

The LIN28 cDNA was cloned into the pMIRNA1 lentiviral vector (System Biosciences, Palo Alto, CA, USA). Transfection efficiencies were confirmed through copGFP. To produce the virus, 293 T cells were plated at 70~80% confluency and treated with plasmids expressing GAG and VSVG using Lipofectamine LTX (ThermoFisher). After 2 days, the supernatant was collected and concentrated using Retro concentin^TM^ (System Biosciences) to infect MSCs.

### 4.13. RNA Sequencing Analysis of miRNAs

For miRNA sequencing, the extracted total RNAs were resolved on a denatured 15% polyacrylamide gel. The gel fragments of 18–26 nucleotides were excised, and small RNAs were eluted overnight with 0.5 M NaCl at 4 °C and precipitated by ethanol. A small RNA sequencing library was generated with 1 μg of RNA using a TruSeq Small RNA Sample Prep Kit version 2 (Illumina, San Diego, CA, USA). Illumina HCS (version 1.4.8), RTA (version 1.12.4.2), and CASAVA (version 1.8.2) software was used for base calling and the generation of raw, de-multiplexed sequencing data in FASTQ format.

### 4.14. Liquid Chromatography and Tandem Mass Spectrometry (LC–MS/MS)

The peptide samples were reconstituted in 0.4% acetic acid and sonicated in a sonication bath at 37 °C for 10 min. The collected protein samples (100 μg protein) were analyzed with nano-flow liquid chromatography tandem mass spectrometry (LC–MS/MS) on an LTQ-Orbitrap XL mass spectrometer (Thermo Fisher Scientific) after tryptic digestion. Mass spectrometric data were loaded onto Proteome Discoverer (ver2.2.0.388) software, and label-free quantification (LFQ) was performed to compare the expression levels of each individual protein. We used the human UniProtKB database (released in June 2020) with the FBS protein list added for the MS data search to exclude suspected proteins affected by FBS. Only the proteins confirmed as true secretory proteins via SignalP, SecretomeP (score ≥ 0.5), or TMHMM were used for the next step of gene ontology (GO) analysis. GO terms were analyzed using the algorithm of the Database for Annotation, Visualization, and Integrated Discovery (DAVID) tools (https://david.ncifcrf.gov/, accessed on 19 March 2021).

### 4.15. Wound-Healing Assay

C57/BL6 mice (8-week-old, male) were obtained from Jackson Laboratories. After shaving the fur, two 8 mm skin wounds were created on each side of the backbone. Each wound received 1 × 10^6^ mMSCs by spraying onto the wound or intradermal injection around the wound. To prevent wound contraction, Tegaderm was placed over the wounds. The wounds were photographed at Days 0, 3, 5, 7, 9, 11. The wound area was measured using the ImageJ analysis program. The percentage of wound closure was calculated as follows: (area of original wound − area of actual wound)/area of original wound × 100.

## 5. Conclusions

We optimized ASOs that could inhibit multiple families of let-7 miRNAs. MSCs inhibited by anti-let-7 ASO exhibited an increment of cell-autonomous functions and stimulating effects on neighboring cells. The ASO against let-7 could be a safe strategy to enhance the therapeutic potential of MSCs.

## Figures and Tables

**Figure 1 ijms-24-08639-f001:**
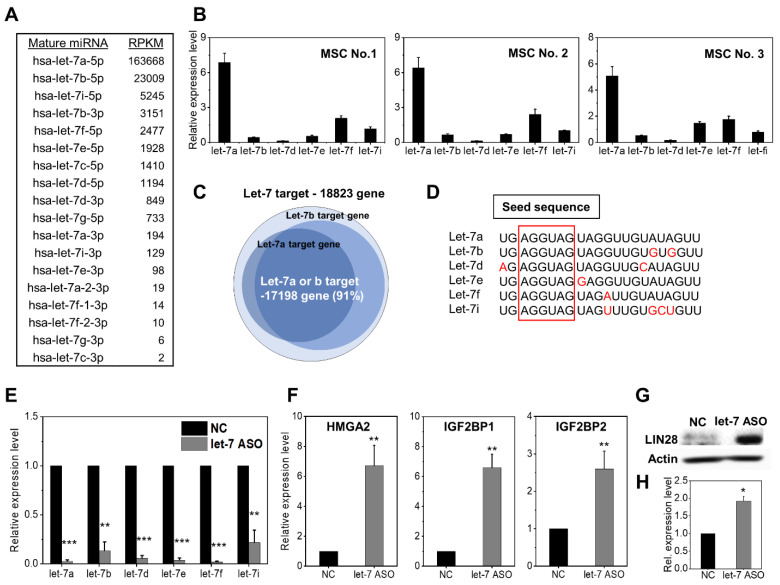
Expression levels of the let-7 miRNA subfamily in mesenchymal stromal cells (MSCs) and their inhibition with selected subsets of antisense oligonucleotides. (**A**) Expression levels of miRNA subfamilies in adipose derived mesenchymal stromal cells (AD-MSC) assessed by reads per kilobase of transcript per million mapped reads (RPKM) in RNA-seq analysis. (**B**) Let-7 expression levels in bone-marrow mesenchymal stromal cells derived from three independent donors. miRNA expression levels were normalized via U6 gene expression levels. (**C**) Numbers of genes targeted by let-7a or let-7b in comparison to the total numbers of target genes of the let-7 family based on bioinformatic analysis with miRWalk DB (http://mirwalk.umm.uni-heidelberg.de/, accessed on 1 March 2023). Note: 91% of total let-7 target genes are covered by let-7a or let-7b. (**D**) Conservation of seed sequences among let-7 family genes. The sequences of each let-7 family gene and its seed sequences are shown boxed with a red line. (**E**) Cross-reactive inhibition of let-7 family genes with selected combinations of ASOs against let-7a and b (anti-let7-ASO). Relative expression levels of each indicated let-7 miRNA family in MSCs were measured in comparison to the levels in cells transfected with scrambled RNA control (NC) via real-time quantitative PCR, normalized by U6. Mean ± SEM is shown (n = 3 each, 3 expts, ** *p* < 0.01; *** *p* < 0.001). (**F**) The induction of let-7 target genes with anti-let7-ASO. Expression levels of each indicated gene were analyzed with RQ-PCR after normalization by HPRT (mean ± SEM, n = 3 each, 3 expts, ** *p* < 0.01). (**G**,**H**) The upregulation of LIN28 proteins with anti-let7-ASO. Western blot image of endogenous LIN28 expression level of let-7-inhibited MSCs. (**G**) Representative Western blot images with (**H**) quantification of protein levels (mean ± SEM, n = 3 each, 3 expts, * *p* < 0.05).

**Figure 2 ijms-24-08639-f002:**
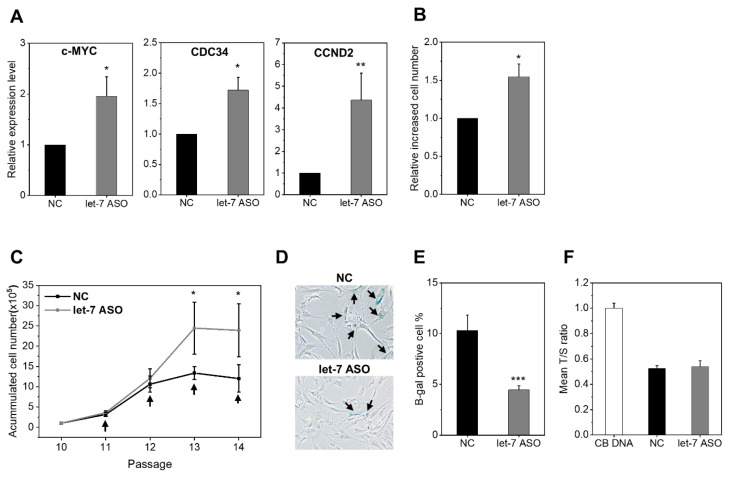
Higher ex vivo expansion of MSCs by anti-let7-ASO. (**A**) Induction of cell-cycle gene transcripts by anti-let7-ASO. Relative expression levels of each indicated cell-cycle gene normalized by HPRT in comparison to a negative control (mean ± SEM, n = 3 each, 3 expts, * *p* < 0.05; ** *p* < 0.01). (**B**) Higher proliferation of MSCs transfected with anti-let7-ASO. MSCs were transfected with anti-let7-ASO, and increases in cell number relative to the negative controls are shown as the mean ± SEM (n = 2 each, 3 expts, * *p* < 0.05). (**C**) Higher expansion capability of MSCs after anti-let7-ASO. MSCs (passage 11) were subjected to long-term expansion culture with the serial transfection of anti-let7-ASO or scrambled control. The accumulated increase in cell number during the passage is shown as measured at each passage subculture. The arrow indicates the transfection of ASO during passaging. Data are the mean ± SEM (n = 2 each, 3 expts, *p* < 0.05)**.** (**D**,**E**) Effects of let-7 inhibition on MSC senescence. MSCs were passaged and stained for senescence-associated β-galactosidase at passage 12. Representative microscopic images (×200) of the (**D**) staining and (**E**) quantification of SA-β-gal-positive cells are shown. Values are mean ± SEM (n = 4 each, 3 expts, *** *p* < 0.001). (**F**) Comparable telomeric length of anti-let7-ASO transfected and control-transfected MSCs. MSCs transfected with anti-let7-ASO or control were cultured for 12 passages, and their telomeric length was measured with RQ-PCR using telomere-specific primers in comparison to the telomeric length in neonatal cord blood cells. The relative telomeric length was determined via the copy numbers of the T (telomere) and S (single copy; b-globin) amounts relative to the 36B4 gene to calculate the relative T/S ratios (n = 2).

**Figure 3 ijms-24-08639-f003:**
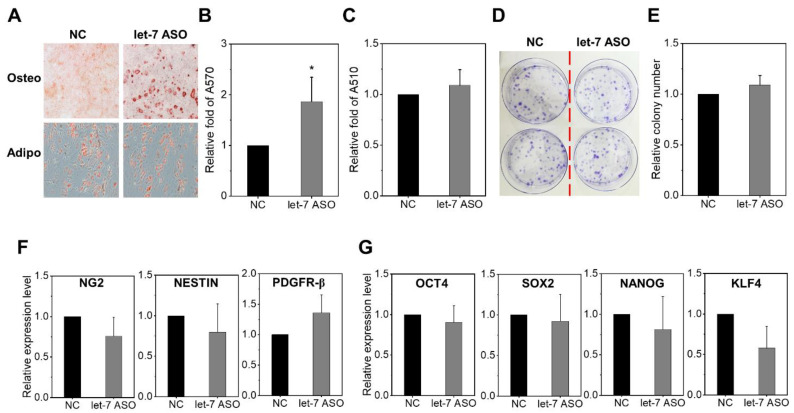
Functional changes of MSCs induced by let-7 inhibition. (**A**–**C**) Effects of let-7 inhibition on MSC multi-lineage differentiation. (**A**) Representative image (×100) of osteogenic or adipogenic differentiation in anti-let7-ASO or control-transfected MSC. (**B**) Quantification of osteogenic differentiation measured by relative mineralization content at A570 nm. Data are the mean ± SEM (n = 3 each, 3 expts, * *p* < 0.05). (**C**) Quantification of adipogenic differentiation measured via relative lipid droplet deposition as determined at A510 nm. Data are the mean ± SEM (n = 3 each, 3 expts). (**D**,**E**) Effects of anti-let7-ASO on cell colony formation. (**D**) Representative images of the colony-forming unit assay (CFU-F) in MSCs and (**E**) relative numbers of CFU-F derived from 2000 MSCs. Values are mean ± SEM (n = 4 each, 3 expts). (**F**,**G**) Effects of anti-let7-ASO on the expression of (**F**) pericyte-marker genes and (**G**) pluripotency-related genes in MSCs. mRNA expression levels were normalized via HPRT. Values are the relative expression levels of each indicated gene in comparison to control MSCs, showing the mean ± SEM (n = 3 each, 3 expts).

**Figure 4 ijms-24-08639-f004:**
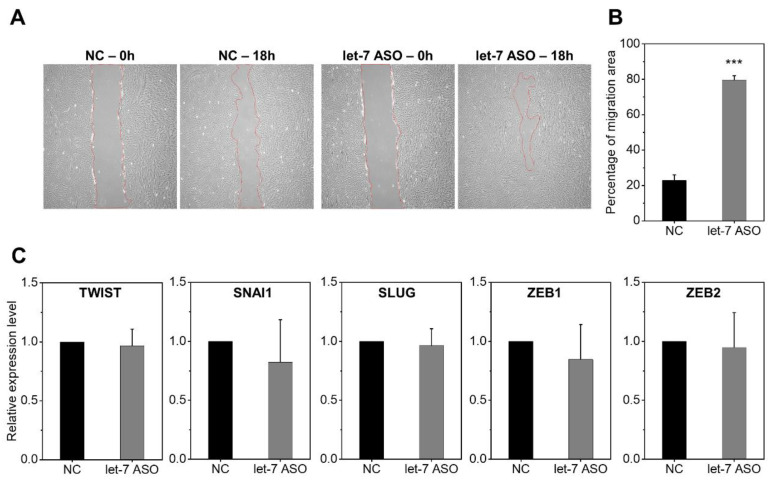
Let-7 inhibition increases MSC cell migration. MSCs were cultured in 6-well plates until confluency, and the culture layers were scratched with a pipette tip to create an area without cells. The migration of MSCs toward the scratched area was measured via ImageJ analysis. Representative images (×50) of cell migration where the edges of cell migration are marked by red dotted lines (**A**) and the migration area is quantified (**B**). The percentage of cell migration was calculated by measuring the differences in the scratched area at 0 and 18 h using the following formula: (scratched area at 18 h − scratched area at 0 h)/initial defective area at 0 h. Mean ± SEM (n = 12 each, 3 expts, *** *p* < 0.001). (**C**) Effects of anti-let7-ASO on the expression of epithelial–mesenchymal transition (EMT)-related genes in MSCs. Each mRNA expression level was normalized by HPRT. Values are the mean ± SEM (n = 3 each, 3 expts) relative expression levels of mRNA.

**Figure 5 ijms-24-08639-f005:**
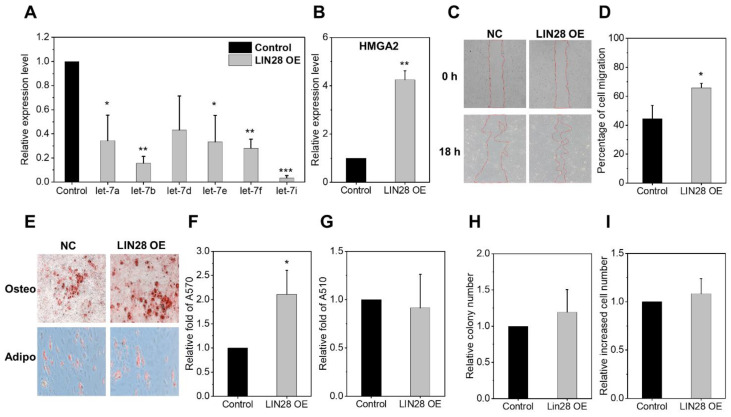
Effects of LIN28 overexpression in MSCs. MSCs were infected with a control vector (pMIR-NA1) or vector-encoding LIN28a to analyze the effects of LIN28 overexpression. (**A**) Downregulation of let-7 expression in LIN28-overexpressed MSCs. Expression levels of each indicated let-7 family gene were analyzed via RQ-PCR after normalization with the U6 gene. Values are the mean ± SEM (n = 3 each, 3 expts, * *p* < 0.05; ** *p* < 0.01; *** *p* < 0.001). (**B**) Induction of let-7 target gene (HMGA2) via the overexpression of LIN28 in MSCs. Values are the relative expression levels of HMGA2 in MSCS (mean ± SEM, n = 3 each, 3 expts, ** *p* < 0.01). (**C**,**D**) Effect of LIN28 overexpression on MSC migration. Representative images (×50) of MSC cell migration in each group are shown with edges marked by red dotted lines (**C**) and the (**D**) quantification of percent migration 18 h after streaking using the following formula: (defective area at 18 h − defective area at 0 h)/defective area at 0 h. The percentage of the migration area with ther mean ± SEM is shown (n = 3 each, 3 expts, * *p* < 0.05). (**E**–**G**) Effects of LIN28 overexpression on multi-lineage differentiation of MSC. (**E**) Representative image (×100) of osteogenic or adipogenic differentiation in LIN28 and control overexpressed MSC. (**F**) Quantification of osteogenic differentiation measured via the relative mineralization content determined at A570 nm. (**G**) Quantification of adipogenic differentiation measured via the relative lipid droplet deposition determined at A510 nm with the mean ± SEM (n = 3 each, 3 expts, * *p* < 0.05) is shown. (**H**) Effects of LIN28 overexpression on MSC colony formation with the mean ± SEM (n = 4 each, 3 expts) relative number of CFU-F derived from 2000 MSCs. (**I**) Effects of LIN28 overexpression on MSC proliferation, showing mean ± SEM (n = 2 each, 3 expts) relative cell numbers compared to controls.

**Figure 6 ijms-24-08639-f006:**
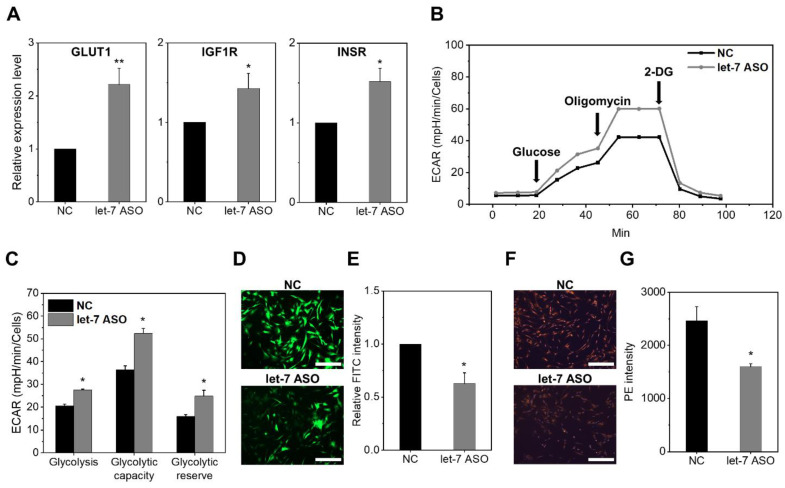
Metabolic changes in MSCs induced by let-7 inhibition. (**A**) Induction of glucose uptake genes via anti-let7-ASO. Expression levels of each indicated let-7 target gene were analyzed with RQ-PCR after normalization by HPRT (mean ± SEM, each n = 3, 3 expts, * *p* < 0.05; ** *p* < 0.01). (**B**,**C**) Comparisons of glycolytic capacity of MSCs assessed by measuring extracellular acidification rate (ECAR). The glycolytic capacity was analyzed by measuring ECAR during the sequential addition of the indicated compounds (indicated by an arrow). (**B**) Representative plots and (**C**) quantifications. Mean ± SD (each n = 3, 1 expts, * *p* < 0.05). (**D**,**E**) Effects of anti-let7-ASO on the generation of ROS. (**D**) Representative images (×180) of MSCs stained by ROS-binding dye (DCF-DA) and (**E**) quantification of the intensity of the stained dye. Images were taken with the same settings (scale bar = 500 µm) with the mean ± SEM (each n = 9, 3 expts, * *p* < 0.05). (**F**,**G**) Effects of anti-let7-ASO on trans-membrane potential in mitochondria. MSCs were stained with TMRE dye to measure the membrane potential of the mitochondria. (**F**) Representative images (×180) of TMRE-stained mitochondria in MSCs and (**G**) quantification of fluorescent intensity in each group of MSCs. Images were taken with the same settings (scale bar = 500 µm). Values are the mean ± SEM (n = 9 each, 3 expts, * *p* < 0.05).

**Figure 7 ijms-24-08639-f007:**
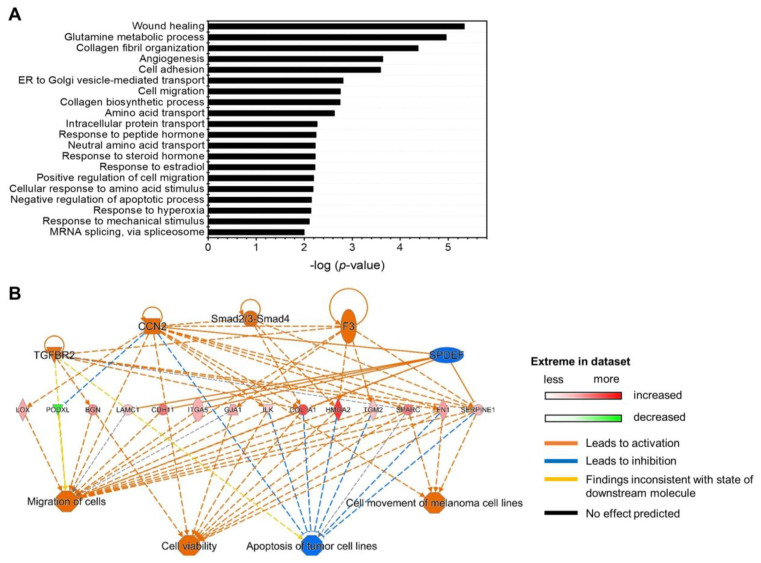
Proteomic analysis of let-7-inhibited MSCs. MSCs transfected with anti-let7-ASO or control ASO were analyzed for proteomic changes via LC-MS/MS at 48 h after transfection. (**A**) Gene ontology analysis for significantly enriched cellular functions in let-7-inhibited MSCs in comparison to control MSCs. Proteins whose expression levels in let-7-inhibited MSCs were higher than those in control MSCs (>1.3 fold) were examined with David analysis (https://david.ncifcrf.gov/home.jsp, accessed on 1 August 2022) (**B**) IPA for changes in cellular functions induced by let-7 inhibition. Predictions of cellular functions induced by differentially expressed proteomes (>1.3 fold) are illustrated with a gene interaction network map. Data were obtained from two independent experiments.

**Figure 8 ijms-24-08639-f008:**
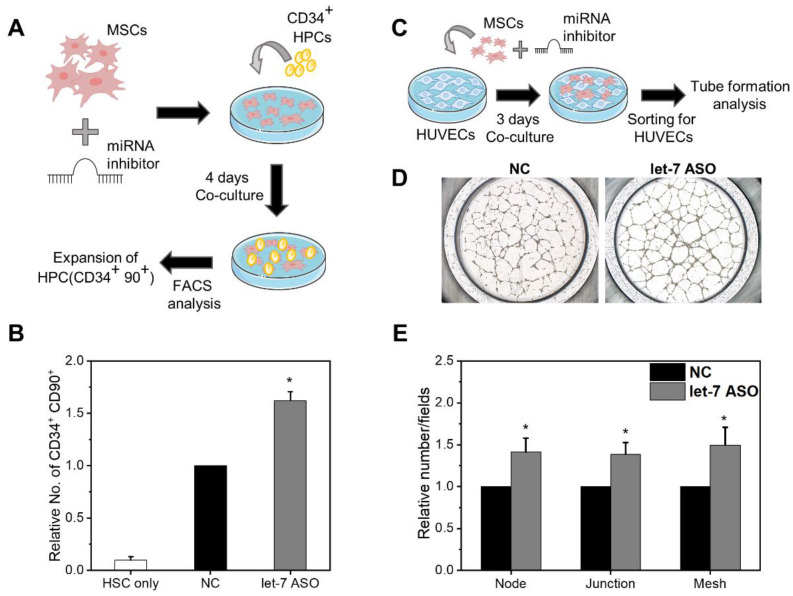
Let-7-inhibited MSCs cause higher stimulating effects on neighboring cells. MSCs transfected with anti-let7-ASO were co-cultured with hematopoietic progenitor cells (HPCs) or human umbilical-vein endothelial cells (HUVEC) to measure their paracrine stimulating effects. (**A**) Schematic of experiments for the co-culture of MSCs and CD34^+^ HPCs. After 4 days of co-culture, the expansion of primitive HPCS was analyzed via flow cytometry to measure the expansion of CD34^+^ 90^+^ subsets. (**B**) Expansion of primitive hematopoietic progenitor (CD34^+^ CD90^+^) cells under stroma-free conditions or co-culture with each group of MSCs. Expansion of CD34^+^ 90^+^ HPCs in let-7-inhibited MSCs relative to the expansion in control MSCs. Mean ± SEM (n = 3 each, 3 expts, * *p* < 0.05). (**C**) Schematic illustration for co-culture of HUVECs and MSCs. After a co-culture with each group of MSCs, HUVECs were sort-purified for tube formation assay on Matrigel. (**D**,**E**) Comparisons of tube formation by HUVECs after co-culture with each group of MSCs. (**D**) Representative images (×25) of tube formation in each group of HUVECs and (**E**) quantification of tube formation as assessed by numbers of nodes, junctions and meshes in the tube. Values are mean ± SEM (n = 5, 3 expts, * *p* < 0.05).

## Data Availability

Not applicable.

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
