# Peer review of "Antisense Oligonucleotides against Let-7 Enhance the Therapeutic Potential of Mesenchymal Stromal Cells"

_ijms, 2023, doi:10.3390/ijms24108639_

Round 1
Reviewer 1 Report
In this manuscript authors shows that antisense oligonucleotides against let-7 enhance the therapeutic potential of mesenchymal stromal cells in detail with some interesting experiments, however, I have some suggestions.
1- in Figure 1A, let-7a-5p was the most abundantly expressed followed by let-7b-5p but in Figure1B expression level of let-7F is higher as compares to 7b. It should be clearly mentioned and explained thoroughly in the text of the manuscript with possible reasons.
2-Figure 1G- Western Blot lanes are not labelled.
3-Figure 4A and all other wound healing/migration images needs to be image and presented at 0 hours as well so it can be seen that initial 0-hour wound was of same size. It's very hard for the readers to understand the presented data without 0-hour images.
4-Figure 6D-F. were these images captured at same acquisition/lens intensity? please mention in the figure legends and text. Also add scale bars to the images.
5-Discussion needs to be written in more detail about what is the impact of this study in aspect of overall knowledge in the field given the fact that let-7, LIn 28 and other targets are previously reported.
Minor editing of English language required.
Author Response
Reviewer 1. Comments and Suggestions for Authors
In this manuscript authors shows that antisense oligonucleotides against let-7 enhance the therapeutic potential of mesenchymal stromal cells in detail with some interesting experiments, however, I have some suggestions.
1- in Figure 1A, let-7a-5p was the most abundantly expressed followed by let-7b-5p but in Figure1B expression level of let-7F is higher as compares to 7b. It should be clearly mentioned and explained thoroughly in the text of the manuscript with possible reasons.
Response) – Yes, we have described the rationale for choosing the combination of the two ASOs.
- The let7a-5p is the miRNA predominantly expressed in MSCs with the rest of let 7 families representing minor populations. While let7b or let7f represent 2nd population in adipose-derived MSCs and bone marrow-derived MSCs, respectively, each of them comprise only minor population.
- The miRNA target DB showed that combination of let-7a and let-7b covers 91% of total let-7 family target genes.
- Most importantly, we showed that ASO combination against let-7a and let-7b could completely suppress the level of let-7f miRNA, too, thus validating the effectiveness of these two combination for most of let-7 family including let-7f.
- We have added these comments on result part (marked in red).
2-Figure 1G- Western Blot lanes are not labelled.
Response) – Yest, we amended the error and labelled the lanes.
3-Figure 4A and all other wound healing/migration images needs to be image and presented at 0 hours as well so it can be seen that initial 0-hour wound was of same size. It's very hard for the readers to understand the presented data without 0-hour images.
Response) – Yes, we have added the initial (day 0) images in Fig 4A and Fig 5C, as suggested.
4-Figure 6D-F. were these images captured at same acquisition/lens intensity? please mention in the figure legends and text. Also add scale bars to the images.
Response) – Yes, we have added the description of the pictures in each figure legend, as suggested.
5-Discussion needs to be written in more detail about what is the impact of this study in aspect of overall knowledge in the field given the fact that let-7, Lin 28 and other targets are previously reported.
Response) – We have added more description on the significance of this study in the light of ASO therapy against let-7 miRNA as a safe strategy to modulate the regenerative activity of MSCs without genetic modification, as well as the strategy for transient stimulation of regenerative microenvironment for tissue regeneration (marked in red).

Reviewer 2 Report
In this study, authors explored the effect of let-7 inhibition on MSCs using antisense oligonucleotides (ASO) and ADMSC as cell model. Their findings showed that the combination of anti-let7-ASO could significantly repress the expression of let-7 family and consequently regulated the let-7 targeted genes including HMGA1, IGF2BP1, and LIN28. By using the anti-let7-ASO, they observed that inhibition of let-7 promoted in vitro MSC proliferation and exhibited higher osteogenic potential and comparable adipogenic differentiation without affecting telomere length in MSCs and frequencies of colony forming potential and stemness-related genes. Meanwhile, this study also showed that let-7 inhibition exhibited similar but not identical effects on MSCs compared to LIN28 overexpression. Moreover, this study also provides evidences that let-7 inhibition MSCs could support hematopoietic progenitor cells (HPCs), but not leukemia cells. In conclusion, the study approach is well designed, their experiments have been properly conducted, and the conclusions can be supported by the results. I think this study has merit and interest, and it expands our knowledge to understand the potential of anti-let7-ASO to promote MSC therapy. However, some important discussions about the opposite results using let-7 inhibitors is missing. Particularly, Sung et al. previously demonstrate that cancer-associated MSCs exhibit loss of let-7 and produce higher level of IL-6 to promote invasiveness of prostate cancer cells, and the similar results are also observed by using let-7 inhibitors (PLoS One 2013, 8(8)e71637). Authors should further discuss these discrepancies.
Author Response
Reviewer 2: Comments and Suggestions for Authors
In this study, authors explored the effect of let-7 inhibition on MSCs using antisense oligonucleotides (ASO) and ADMSC as cell model. Their findings showed that the combination of anti-let7-ASO could significantly repress the expression of let-7 family and consequently regulated the let-7 targeted genes including HMGA1, IGF2BP1, and LIN28. By using the anti-let7-ASO, they observed that inhibition of let-7 promoted in vitro MSC proliferation and exhibited higher osteogenic potential and comparable adipogenic differentiation without affecting telomere length in MSCs and frequencies of colony forming potential and stemness-related genes. Meanwhile, this study also showed that let-7 inhibition exhibited similar but not identical effects on MSCs compared to LIN28 overexpression. Moreover, this study also provides evidences that let-7 inhibition MSCs could support hematopoietic progenitor cells (HPCs), but not leukemia cells. In conclusion, the study approach is well designed, their experiments have been properly conducted, and the conclusions can be supported by the results. I think this study has merit and interest, and it expands our knowledge to understand the potential of anti-let7-ASO to promote MSC therapy. However, some important discussions about the opposite results using let-7 inhibitors is missing. Particularly, Sung et al. previously demonstrate that cancer-associated MSCs exhibit loss of let-7 and produce higher level of IL-6 to promote invasiveness of prostate cancer cells, and the similar results are also observed by using let-7 inhibitors (PLoS One 2013, 8(8)e71637). Authors should further discuss these discrepancies.
Response) - We appreciate this important point by the reviewer. We have added comments on potential to stimulate cancer cells as well as normal progenitor cells along with the references (marked in red).

Round 2
Reviewer 1 Report
In this revised version, all the comments has been addressed and manuscript is improved in various aspects.
Reviewer 2 Report
The previous concerns have been properly addressed. In addition, discussion of the different roles of let-7 miRNA in tumor cells has also been significantly improved.